# SCN1A—Characterization of the Gene's Variants in the Polish Cohort of Patients with Dravet Syndrome: One Center Experience

Elżbieta Stawicka [1,*], Anita Zielińska [2,*], Paulina Górka-Skoczylas [2] , Karolina Kanabus [2], Renata Tataj [2], Tomasz Mazurczak [1] and Dorota Hoffman-Zacharska [2]

[1] Clinic of Paediatric Neurology, Institute of Mother and Child, Kasprzaka 17A, 01-211 Warsaw, Poland; tomasz.mazurczak@imid.med.pl

[2] Department of Medical Genetics, Institute of Mother and Child, Kasprzaka 17A, 01-211 Warsaw, Poland; paulina.gorka@imid.med.pl (P.G.-S.); karolina.kanabus@imid.med.pl (K.K.); renata.tataj@imid.med.pl (R.T.); dorota.hoffman@imid.med.pl (D.H.-Z.)

[*] Correspondence: elzbieta.stawicka@imid.med.pl (E.S.); anilewandowska@imid.med.pl (A.Z.); Tel.: +48-792-062-919 (A.Z.)

**Abstract:** The aim of this study was to characterize the genotype and phenotype heterogeneity of patients with *SCN1A* gene mutations in the Polish population, fulfilling the criteria for the diagnosis of Dravet syndrome (DRVT). Particularly important was the analysis of the clinical course, the type of epileptic seizures and the co-occurrence of additional features such as intellectual disability, autism or neurological symptoms such as ataxia or gait disturbances. Based on their results and the available literature, the authors discuss potential predictors for DRVT. Identifying these early symptoms has important clinical significance, affecting the course and disease prognosis. 50 patients of the Pediatric Neurology Clinic of the Institute of Mother and Child in Warsaw clinically diagnosed with DRVT and carriers of *SCN1A* pathogenic variants were included. Clinical data were retrospectively collected from caregivers and available medical records. Patients in the study group did not differ significantly in parameters such as type of first seizure and typical epileptic seizures from those described in other studies. The age of onset of the first epileptic seizure was 2–9 months. The co-occurrence of intellectual disability was confirmed in 71% of patients and autism in 18%. The study did not show a correlation between genotype and phenotype, considering the severity of the disease course, clinical symptoms, response to treatment, the presence of intellectual disability, autism symptoms or ataxia. From the clinical course, a significant problem was the differentiation between complex febrile convulsions and symptoms of DRVT. The authors suggest that parameters such as the age of the first seizure, less than one year of age, the onset of a seizure up to 72 h after vaccination and the presence of more than two features of complex febrile seizures are more typical of DRVT, which should translate into adequate diagnostic and clinical management. The substantial decrease in the age of genetic verification of the diagnosis, as well as the decline in the use of sodium channel inhibitors, underscores the growing attention of pediatric neurologists in Poland to the diagnosis of DRVT.

**Keywords:** *SCN1A* gene; Dravet syndrome; DRVT; genotype–phenotype correlation



## 1. Introduction

Since the initial discovery of the *SCN1A* gene (OMIM 182389)'s association with epilepsy by Escayg et al. [1], it consequently remains the most common and essential epilepsy pathogenic gene. Situated on chromosome 2 at locus 2q24.3, the *SCN1A* gene encodes the leading functional alfa subunit of one of the voltage-gated sodium channels (NaVs)—Nav1.1 [2]. These channels play a pivotal role in regulating cellular excitability, governing the initiation and propagation of action potentials in various excitable

cells, including nerves, muscles and neuroendocrine cells. Nav1.1 channels are predominantly found within GABA-ergic interneurons across many central nervous system (CNS) regions [3]. The inhibitory function of these interneurons contributes to maintaining a balance between excitation and inhibition in the CNS [4]. Disruption of this balance, in the case of *SCN1A* gene mutations, due to the restricted function of interneurons results in clinical symptoms that can manifest as both epileptic (a heterogeneous group of epileptic syndromes) and non-epileptic. Epileptic syndromes include disorders ranging in severity from relatively mild familial febrile seizures (FEB3A; 604403), Genetic Epilepsy with Febrile Seizure Plus (GEFSP2/GEFS+2; OMIM 604403), to the much more severe prognostically units, categorized as developmental and epileptic encephalopathy (DEE), such as Dravet syndrome (DRVT; OMIM 607208) or infantile developmental and epileptic encephalopathy 6B, non-Dravet type (DEE6B; OMIM 619317). Among non-epileptic disorders, familial hemiplegic migraine type 3 (FHM3; OMIM 609634) has been described to date. Seldom, *SCN1A* variants are also associated with other forms of DEE, including myoclonic–astatic epilepsy (MAE, Doose syndrome), Infantile epilepsy with migratory focal seizures (EIMFS), West syndrome (WS) and Lennox–Gastaut syndrome (LGS) [5].

Mutations in the *SCN1A* gene represent the most commonly identified cause of developmental and epileptic encephalopathies (DEEs) also among Polish patients [6–8]. Consequently, *SCN1A* is the most extensively studied gene concerning the characteristics and pathogenicity of identified variants. More than 2600 different mutations have been recently described in the *SCN1A* gene, of which 2346 have a pathogenic status and about 300 are variants of unknown significance (VUS) (HGMD Professional v.2023.3; https://my.qiagendigitalinsights.com/bbp/view/hgmd, accessed on 30 December 2023). The vast majority (>80%) of the described pathogenic variants underlie DRTV and are predominantly de novo mutations. Pathogenic variants in the *SCN1A* gene are of a dominant character; in DRTV and GEFSP2, syndromes lead to loss of protein function (loss-of-function; LOF). Pathogenic variants in FHM3 and non-Dravet DEE syndromes are gain of function (GOF) in nature.

DRVT represents a severe, drug-resistant DEE. According to estimates, the syndrome's incidence ranges from 1:15,000 to 1:40,000. Seizures that are provoked by hyperthermia (fever, vaccinations and hot baths) with polymorphic morphology (clonic hemispheric, generalized and/or tonic–clonic seizures) typically appear in the first phase of the disease—on average at six months of age. Patients' psychomotor development is expected until the onset of the first seizures. In the second phase, between 1 and 4 years of age, new types of seizures are observed (most often myoclonic, atypical absent, atonic, often prolonged to status epilepticus), regression of psychomotor development with increasing symptoms of autism spectrum disorders and cerebellar signs reveals itself. The third phase, which usually begins after the age of 5, is a period of decline in the severity of epileptic seizures (tonic–clonic, focal clonic seizures, myoclonic seizures are less common) and increasing symptoms of ataxia. In addition, pyramidal and extrapyramidal signs (parkinsonism, chorea, choreoathetosis) and cognitive dysfunction appear. DRVT is characterized by high drug resistance, a notable risk of status epilepticus and sudden unexpected death in epileptic patients (SUDEP) [9].

The genotype–phenotype correlation of *SCN1A*-related seizure disorders still remains uncertain; however, recent data from extensive cohort studies seem to be giving us some clues [10]. Our study retrospectively documented the clinical phenotype and genotype data of 50 patients diagnosed in one center with DRVT due to *SCN1A* variants. The aim of the study was to explore genotype–phenotypic associations in patients with DRVT, as well as the analysis of the implementation of bioinformatic tools and databases of molecular data in the diagnostic process. The results aim to define the phenotypic spectrum and provide a scientific basis for treatment and genetic counseling.

## 2. Materials and Methods

### 2.1. Patients' Clinical Characterization

The retrospective analysis encompassed 50 patients whose genetic testing confirmed the presence of pathogenic/likely pathogenic variants in the *SCN1A* gene. Comprehensive clinical data for each patient were considered, incorporating details such as the seizure types, age at onset, frequency and duration of seizures, as well as concurrence of other features such as intellectual disability, autistic signs and neurological deficits. The pediatric neurologist conducted the classification of epileptic seizures according to the current diagnostic criteria of the International League Against Epilepsy. Patients were confirmed with a diagnosis of Dravet syndrome based on the following criteria (according to ILAE 2022 [11]):

1. Onset of seizures typically between 3 and 9 months, in rare cases 1–20 months.
2. Standard head size during the first years of life.
3. Intellectual disability or regression of psychomotor development.
4. Type of seizures: recurrent focal clonic (hemiclonic) febrile and afebrile seizures (which often alternate sides from seizure to seizure), focal to bilateral tonic–clonic and/or generalized clonic seizures.
5. Additional seizure types (not mandatory): myoclonic seizures, focal impaired awareness seizures, focal to the bilateral tonic–clonic seizures, atypical absence seizures, atonic seizures, nonconvulsive status epilepticus, tonic and tonic–clonic seizures mainly in sleep and in clusters.
6. Drug-resistant epilepsy.
7. Course of illness: drug-resistant seizures with potential episodes of status epilepticus.

The exclusion criteria were as follows:

1. Epileptic spasms.
2. Early infantile *SCN1A* DEE.
3. No history of prolonged seizures (>10 min).
4. Lack of fever sensitivity as a seizure trigger.
5. Typical EEG background without interictal discharges after age two years.
6. Focal neurological findings.
7. MRI showing a causal focal lesion: brain malformation, hypoxic-ischemic brain injury, brain tumor, neurocutaneous disorders, etc.
8. First seizure diagnosis during the neonatal period.

The motor impairment was ranked as unaffected, with mild gait imbalance and ataxic. The intellectual potential was examined by neuropsychologists on the basis of adequate scales (Leiter International Performance Scale, the Brunet–Lézine Scale or the Wechsler Intelligence Scale for Children). Patients with a diagnosis of autism were reanalyzed for clinical symptoms by a child and adolescent psychiatrist. It was carried out based on an interview with parents, psychiatric examination, analysis of available documentation, a diagnostic questionnaire, The Autism Diagnostic Interview-Revised and ICD-10 criteria.

### 2.2. SCN1A Variants Analysis

The analysis of the *SCN1A* gene for all 50 subjects involved sequencing for point mutations. This sequencing was conducted through the direct sequencing of 26 coding exons using the Sanger method (BigDye Terminator v3.1 Cycle Sequencing Kit; ABI PRISM® 3700 Genetic Analyzer, Thermo Fisher Scientific Inc., Waltham, MA, USA). Additionally, targeted next-generation sequencing (tNGS) was employed for DEE-related genes, using EIEE v.1/2016, or NBE v.1/2018 panels (http://zgm.imid.med.pl/panele-ngs/, accessed on 30 December 2023). Sequencing was performed using the Illumina MiSeq platform. In parallel, multiplex ligation-dependent probe amplification (MLPA) was utilized to analyze deletion/duplication in the *SCN1A* gene (SALSA MLPA kit P137; MRC Holland, Amsterdam, Netherlands). Whenever feasible, an analysis of the inheritance pattern of identified variants was performed. This involved using an appropriate method based on the nature of the mutation, utilizing either Sanger sequencing or MLPA. Sanger sequencing was per-

formed for most probands—44, targeted NGS for 6 probands, MLPA for all probands with no point mutations identified, and all with new (not reported as pathogenic in databases) missense variants or in-frame duplication.

The pathogenicity of the identified variants was assessed based on classification data according to ACMG [12], the human pathogenic mutation databases HGMD Professional 2023.3, ClinVar (https://www.ncbi.nlm.nih.gov/clinvar/, accessed on 30 December 2023) and explanatoriness in a clinical context (http://epilepsygenetics.net/2023/02/08, accessed on 30 December 2023).

Bioinformatics tools were used to assess the potential pathogenicity of identified missense variants' function (https://funnc.shinyapps.io/shinyappweb, accessed on 29 January 2024), splicing variants MobiDetails (https://mobidetails.iurc.montp.inserm.fr/MD/, accessed on 29 January 2024).

*SCN1A* gene-dedicated databases SCNportal (https://scn-portal.broadinstitute.org/, accessed on 29 January 2024) and *SCN1A* mutation database (http://SCN1A.caae.org.cn/, accessed on 29 January 2024) were analyzed for information on previously published mutations and their characteristics.

The variants identified in this study are described according to HGVS v.21.0.1 nomenclature recommendations (http://varnomen.hgvs.org/recommendations, accessed on 29 January 2024) based on *SCN1A* reference sequences NM_001165963.4, NG_011906 (GRCh38).

## 3. Results

### 3.1. Clinical Features of Patients

The study group (N = 50) consisted of patients aged 3–30 years (2023 data), with a mean age of 15 years (SD = 6.7). The M/F ratio was 25/25.

According to the obtained data, the parents were not related in any family.

Results for the whole group: The course of pregnancy was significantly complicated in three patients (eclampsia, pre-eclampsia, cervical insufficiency). Delivery was at 34 Hbd in two patients snf in one at 27 Hbd in the remaining patients at term. No microcephaly was observed in neonatal measurements. Head circumference measured at birth in all patients ranged between the 10 and 90th percentile.

Except for one patient (born at 27 Hbd), no significant complications of the perinatal period were observed. The patient born at 27 Hbd received a low Apgar score (3–5), required respiratory support and was diagnosed with RDS (respiratory distress syndrome), ROP (Retinopathy of Prematurity) and grade 1 IVH (Intraventricular Hemorrhage). Family history was burdened with the occurrence of febrile seizures in first-degree relatives in three patients and epilepsy in two.

The mortality rate in the study group was zero.

The 43 patients initially presented seizures provoked by hyperthermia. In five, it was related to vaccination; these patients did not require hospitalization, and they were diagnosed with post-vaccination reactions. The remaining group of 38 patients were hospitalized; 36 of them were diagnosed with simple febrile seizures and were discharged without the inclusion of anticonvulsant drugs permanently. In two patients, due to the onset of status epilepticus and recurrence of seizures within and after 24 h, epilepsy was diagnosed, and treatment with valproic acid was initiated. The patients' clinical data are summarized in Table 1, which contains general characteristics, and Table 2, which contains seizure data.

**Table 1.** Summary of patients' clinical data.

|  | N = 50 |
| --- | --- |
| Mean age in 2023 | 14.9 (3–30 years) |
| The mean age of the first symptoms | 5.3 (2–9 months) |

**Table 1.** *Cont.*

|  | N = 50 |
|---|---|
| Type of the first epileptic seizure: |  |
| Focal clonic/hemiclonic | 53% |
| Tonic–clonic | 39% |
| Convulsive status epilepticus (tonic–clonic) | 6% |
| Tonic during sleep | 2% |
| Neurodevelopmental disorders: |  |
| Autism | 18% |
| ADHD | 10% |
| Intellectual disability—total | 71% |
| mild | 22% |
| moderate | 32% |
| severe | 12% |

**Table 2.** Summary of patients' seizure data.

| Types of Seizures | N = 50 |
|---|---|
| Focal clonic | 80% |
| Hemiclonic | 8% |
| Focal clonic generalized to tonic–clonic | 7% |
| Mioclonic | 55% |
| Atypic absence | 40% |
| Tonic–clonic | 60% |
| Focal tonic | 15% |
| Status epilepticus | 100% |
| convulsive | 82% |
| nonconvulsive | 26% |
| Triggers: |  |
| Hyperthermia | 100% |
| During Infection | 90% |
| After vaccination | 16% |
| High ambient temperature/overheating/hot water | 12% |
| Infection without fever | 22% |
| Emotional | 4% |

### 3.2. Molecular Features of Patients

Here, we exclusively focus on patients whose variants in the *SCN1A* gene were identified as either the confirmed cause or a potential cause (pathogenic/likely pathogenic variants) of DRVT. Within the analyzed group, a total of 49 unique variants were described (private), with only two-point mutations p.Arg712* and p.Lys1846Serfs*11 identified twice. *SCN1A* deletions were described in two cases encompassing all exons and, in one case, eight exons in the 3′ part of the gene. Among identified variants, 39% were novel and not previously documented in pathogenic variants databases (Table 3).

Two out of fifty probands carried two mutations in the *SCN1A* gene (S_19, S_51). In the case of S_19, the de novo nonsense mutation p.Tyr1075* coexisted with the missense variant p.Tyr1598Phe inherited from the unaffected patient's mother (the allelic status of this variant was undetermined). In S_51, de novo p.Arg712* was allelic with p.Thr1174Ser of paternal origin. In both cases, nonsense mutations were considered causative (Table 3).

The identified variants were distributed along the entire gene, with no observed mutation hotspots (Figure 1).

**Table 3.** Genetic and pathogenetic characteristics of the variants identified in 50 DRVT patients included in this study.

| | c.DNA (NM_001165963.4) | Protein | Inheritance | Nav1.1 Localization | ACMG Classification | ClinVar | HGMD | Pathogenicity Classification |
|---|---|---|---|---|---|---|---|---|
| S_1 | 235A>G | Asp79Gly | de novo | N-ter | LPat | Pat (EIEE) | -; Asp79Asn—NDD/DRVT | Pat |
| S_2 | 241G>A | Asp81Asn | de novo | N-ter | LPat | - | + DRVT | Pat |
| S_3 | 278T>A | Leu93* | de novo | N-ter | Lpat | - | - | Pat |
| S_4 | 298T>A | Phe100Ile | nd | N-ter | LPat | - | -; Phe100.Val—DRVT/EOE/GEFS+ | Pat |
| S_5 | 302G>A | Arg101Gln | de novo | N-ter | LPat | Pat (EIEE) | + DRVT | Pat |
| S_6 | 429_430delGT | Val143Tyrfs*148 | nd | DI-S1 | Pat | Pat (DRVT/EIEE) | + DRVT | Pat |
| S_7 | 680T>G | Ile227Ser | de novo | DI-S4 | Pat | Pat (DRVT) | + DRVT | Pat |
| S_8 | 686delT | Val229Alafs*5 | de novo | DI-S4 | LPat | - | - | Pat |
| S_9 | 773T>C | Leu258Pro | nd (ma. -/-) | DI-S5 | LPat | - | - | LPat |
| S_10 | 1025C>T | Ala342Val | de novo | Exter (Loop, IS5-S6) | Pat | Pat (DRVT) | + DRVT | Pat |
| S_11 | 1247A>G | Asn416Ser | de novo | DI-S6 | Pat | LPat (DRVT) | + DRVT | Pat |
| S_12 | 1738C>T | Arg580* | de novo | L1 | Pat | Pat (DRVT/EIEE) | + DRVT | Pat |
| S_13 | 1837C>T | Arg613* | de novo | L1 | Pat | Pat (DRVT/EIEE) | + DRVT | Pat |
| S_14 | 2134C>T | Arg712* | de novo | L1 | Pat | Pat (DRVT/EIEE) | + DRVT | Pat |
| S_51 | 2134C>T | Arg712* | de novo | L1 | Pat | Pat (DRVT/EIEE) | + DRVT | Pat |
| S_15 | 2420dupT | Thr808Hisfs*29 | de novo | DII-S1 | Pat | Pat (nd) | - | Pat |
| S_16 | 2585G>A | Arg862Gln | de novo | DII-S4 | Pat | Pat (DRVT/EIEE) | + VUS DRVT/Epi | Pat |
| S_42 | 2692_2706dupGCCATCATCGTCTTC | Ala898_Phe902dup | de novo | DII-S5 | LPat | - | - | Pat |
| S_17 | 2791C>T | Arg931Cys | de novo | Exter (Loop, IIS5-S6) | LPat | Pat (nd) | + DRVT | Pat |
| S_18 | 2837G>A | Arg946His | de novo | Exter (pore-forming) | LPat | Pat (DRVT/EIEE) | + NDD; DRVT, GEFS+ | Pat |
| S_19 | 3225T>A(;)4793A>T | Tyr1075*(;)Tyr1598Phe | de novo; ma. | L2 | LPat; LPat | -; VUS | -; + (Schiz + Aut, GEFS+, DRVT) | Pat/LPat |
| S_21 | 3734_3735ins TGATCAGC | Lys1246Aspfs*27 | pat moz | Exter(Loop) | Pat | - | - | Pat |
| S_22 | 4168G>A | Val1390Met | de novo | Exter (Loop IIIS5-S6) | Pat | Pat (DRVT/EIEE) | + DRVT | Pat |
| S_23 | 4274T>A | Leu1425* | de novo | Exter (pore-forming) | LPat | - | - | Pat |
| S_24 | 4388T>C | Phe1463Ser | ma. moz | DIII-S6 | LPat | Pat (DRVT) | + DRVT | Pat |
| S_25 | 4459_4460del | Asn1487Profs*22 | de novo | L3 | LPat | - | - | Pat |
| S_26 | 4532T>G | Met1511Arg | nd | L3 | LPat | LPat (nd) | + DRVT/NDD | Pat |
| S_27 | 4539delA | Lys1513Asnfs*2 | de novo | L3 | LPat | - | - | Pat |
| S_28 | 4547C>A | Ser1516* | de novo | L3 | Pat | Pat (DRVT/EIEE/HM) | + DRVT | Pat |
| S_29 | 4783_4784delCT | Leu1595Thrfs*13 | de novo | Cyto (Loop IV S2-S3) | Pat | Pat/LPat (AD EPI/nd) | - | Pat |
| S_30 | 4787G>A | Arg1596His | pat (epi fam) | Cyto (Loop IV S2-S3) | LPat | ConfIntPat, Pat (DRVT/EIEE) Lpat (-) VUS (-) | + GEFS+ (VUS EIEE) | Pat VUS (DRVT) |
| S_31 | 4786C>T | Arg1596Cys | pat (epi fam) | Cyto (Loop IV S2-S3) | Pat | Pat/LPat (FE/GEFS+/EIEE) | + FE/GEFS+/DRVT | Pat VUS (DRVT) |
| S_32 | 4906C>T | Arg1636* | de novo | DIV-S4 | Pat | Pat (DEE6B/DRVT/GEFS+) | + DRVT | Pat |
| S_33 | 4964G>T | Gly1655Val | de novo | Cyto (Loop IV S4-S5) | LPat | - | - p.Gly1655Ala—Epi | Pat |
| S_34 | 5107G>T | Asp1703Tyr | de novo | Exter (Loop IVS5-S6) | LPat | - | + DRVT | Pat |
| S_35 | 5129T>C | Phe1710Ser | de novo | Exter (poreforming) | LPat | - | + VUS DRVT | Pat |
| S_36 | 5178G>A | Trp1726* | de novo | Exter (poreforming) | Lpat | - | + DRVT | Pat |

**Table 3.** *Cont.*

| | c.DNA (NM_001165963.4) | Protein | Inheritance | Nav1.1 Localization | ACMG Classification | ClinVar | HGMD | Pathogenicity Classification |
|---|---|---|---|---|---|---|---|---|
| S_50 | 5383G>T | Glu1795* | pa moz | C-ter | LPat | Pat (EIEE) | - | Pat |
| S_37 | 5432T>A | Val1811Asp | nd | C-ter | LPat | - | + CAE | LPat |
| S_38 | 5536_5539delAAAC | Lys1846Serfs*11 | de novo | C-ter | Pat | Pat (DEE6B/DRVT/GEFS+) | + DRVT/GEFS+/ASD | Pat |
| S_49 | 5536_5539delAAAC | Lys1846Serfs*11 | de novo | C-ter | Pat | Pat (DEE6B/DRVT/GEFS+) | + DRVT/GEFS+/ASD | Pat |
| S_39 | 5734C>T | Arg1912* | de novo | C-ter | LPat | Pat (HM, AD Epi, EIEE) | + DRVT/GEFS+, FS+ | Pat |
| S_40 | 5779A>G | Arg1927Gly | de novo | C-ter | LPat | - | + ASD/NDD | LPat |
| S_20 | 3421_3429+7del | p.? | nd | | LPat | - | - | Pat |
| S_43 | 2589+2dupT | p.? | de novo | | LPat | LPat (DEE6B/DRVT/GEFS+/HM) | + DRVT (c.2589+3A>T) | Pat |
| S_44 | Ex19_26del | P.? | nd (pa. -/-) | | Pat | Pat (DRVT/EIEE) | + del DRVT | Pat |
| S_45 | 2947-1G>A | p.? | de novo | | LPat | Pat (-) | + DRVT | Pat |
| S_46 | Ex1_26 del | - | de novo | | Pat | Pat (DRVT/EIEE) | + del DRVT | Pat |
| S_47 | 4338+1G>A | p.? | nd | | LPat | Pat (EIEE) | + DRVT | Pat |
| S_48 | Ex1_26del | - | de novo | | Pat | Pat (DRVT/EIEE) | + DRVT | Pat |

ma—maternal, pa—paternal, moz—mosaicism, nd—no data, Pat—pathogenic, LPat—likely pathogenic, VUS—variants on unknown significance, confli, EIEE—early infantile epileptic encephalopathy, DRVT—Dravet syndrome, HM—hemiplegic migraine, FS+—febrile seizures plus, GEFS+—generalized epilepsy with febrile seizures plus, ASD—autism spectrum disorder, NDD—neurodevelopmental disorder.

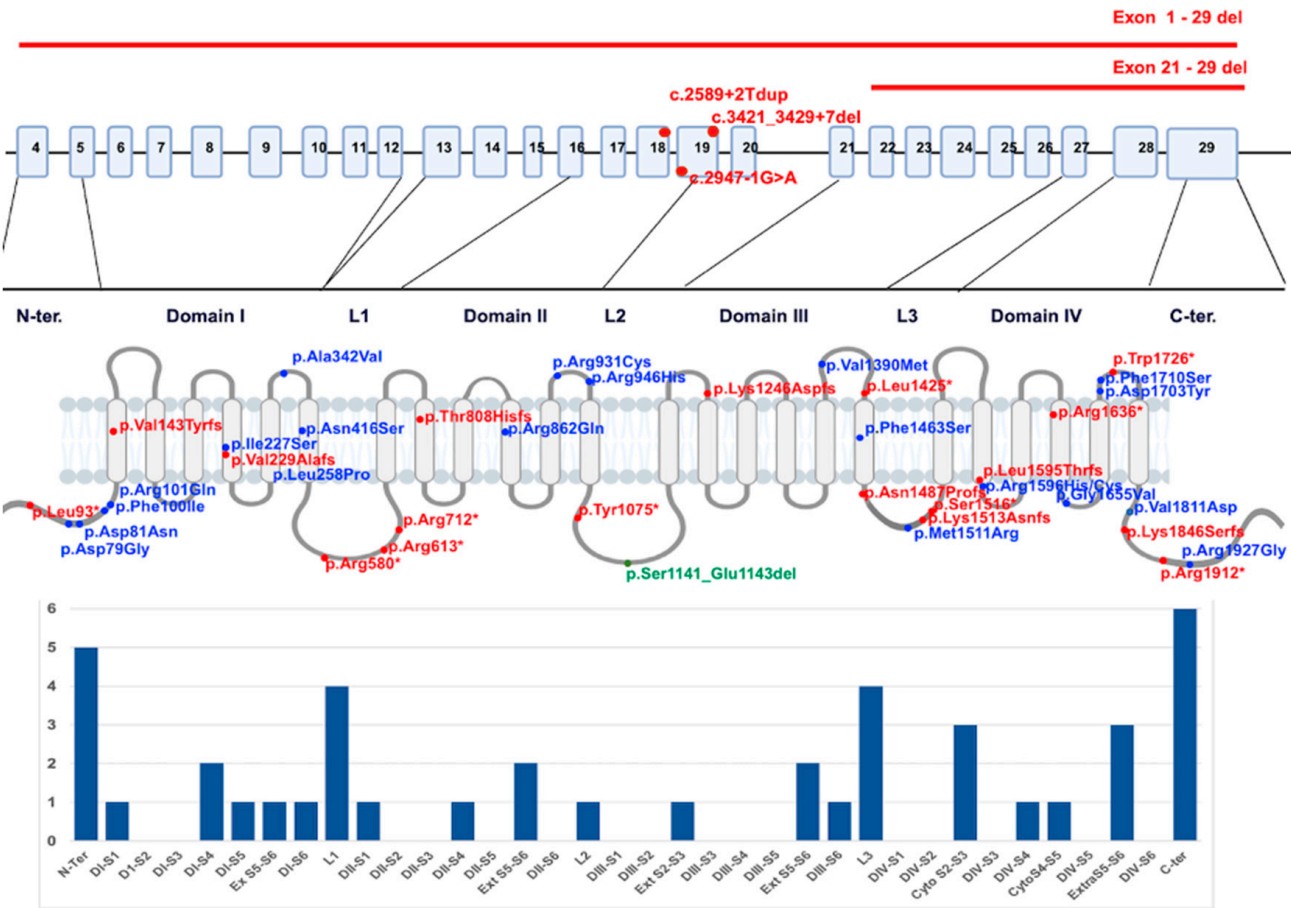

**Figure 1.** Distribution of identified variants along the *SCN1A* gene's coding exons (splicing variants, deletions) and Nav1.1 protein (missense, stop, frameshift and in frame). The alpha subunit consists of four homologous domains (D1–D4), each formed of six transmembrane segments (S1–S6). Segment 4 represents the voltage sensor, and S5–S6 represents the pore region. Individual variants are displayed in different colors; truncating variants are marked in red, missense blue and in frame dup in green.

Truncating mutations were the most common at 56%, including 22% nonsense mutations, 20% frameshift mutations, 6% exon deletions and 8% splicing mutations. Missense mutations constituted 42% of identified variants (Figure 2A). The origin of the mutations was characterized in 86% of patients. The majority were de novo 75%. Paternal inheritance was observed in two cases (4%), associated with the occurrence of the familial form of the disease and intra-familial evolution of phenotype (S_30, S_31) in one family of affected siblings (S_4) due to paternal mosaicism were observed. Additionally, the inheritance of pathogenic variants was observed in patients showing somatic mosaicism—in 6% of cases (more common in paternal—4%, than in maternal 2%). In one case (S_50), the level of nonsense variant in paternal leukocytes' DNA was as high as heterozygous mutations—50:50, but in material isolated from cells of ectodermal origin was lower, at about 20%. In other cases (S_21, S_24), mosaicism was identified in blood at a low level (<20%). All examined parents were healthy. In 13% of patients, we could not perform inheritance analysis; however, in two cases, parental origin of mutation was excluded in one of the parents (S_9, S_44) (Figure 2B).

Genetic characteristics of *SCN1A* variants identified in this study are presented in Table 3. In the analysis, particular attention was paid to missense variants, as they constitute the most significant interpretative problem in terms of pathogenicity, and additionally, the distinction between the functional disorders they cause (LOF vs. GOF) may be necessary for the therapeutic management of patients. The 21 different missense variants were identified in the group under analysis. Their potential pathogenicity was an-

alyzed using several bioinformatics tools. One of them was a machine learning method that also predicts functional effects of genetic missense variants function. It provides two predictions, pathogenic versus neutral and LOF vs. GOV; in 11 cases, variants were predicted as pathogenic and LOF, which is characteristic of DRVT. Among them, three were previously functionally studied and characterized as LOF (p.Ile227Ser [13]; p.Arg946His [14] and p.Arg1927Gly [15]). In six cases, variants were classified as probably neutral, and therefore, the functional prediction is unreliable (but the probability to be LOF). For one of those variants, p.Arg1596Cys, functional data were available indicating LOF [16] or partial LOF (pLOF) [17]. However, in four cases, variants were predicted as pathogenic and GOF; patients did not differ significantly I n the severity of their disease course, clinical symptoms, response to treatment, the presence of intellectual disability, autism symptoms or ataxia. Characteristics of identified missense variants are summarized in Table 4.

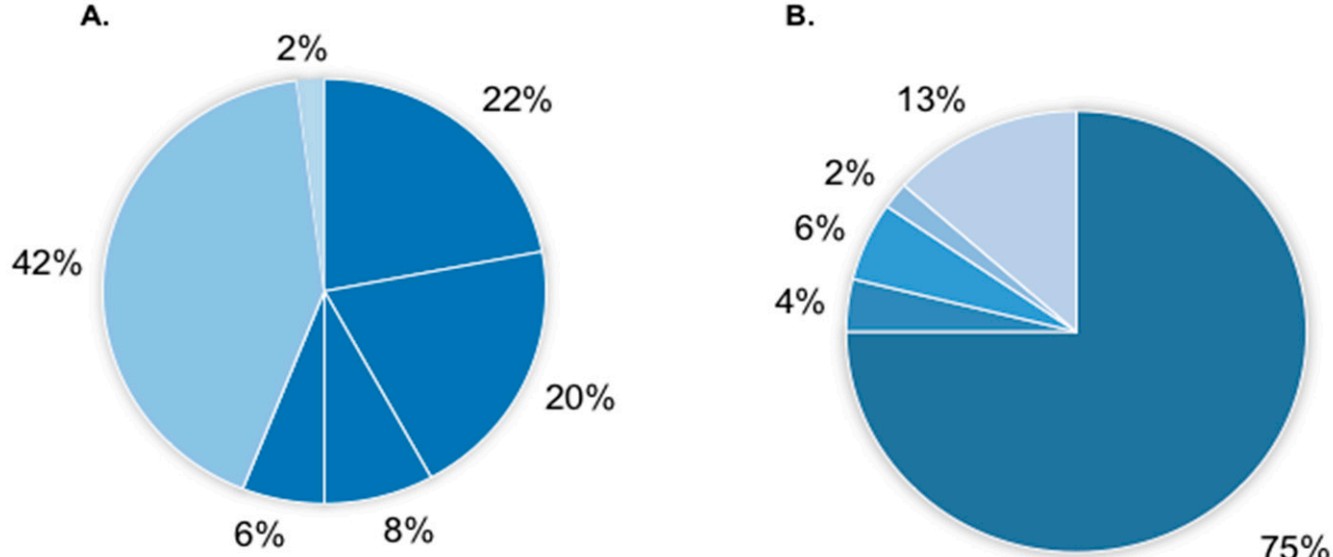

**Figure 2.** Frequency of identified *SCN1A* gene variants in the group of analyzed 50 patients with DRTV according to the type of variant (**A**) and inheritance (**B**). (**A**)—54% protein truncating variants (22% stop, 20% frameshift, 8% splicing and 6% gene/exons deletions), 42% missense variants, 2% in frame del/dup. (**B**)—75% de novo variants, 8% inherited (2% of paternal origin, 10% from mosaic parents—6% paternal, 4% maternal) and 13% without data.

**Table 4.** Characteristics of *SCN1A* gene missense variants identified among patients in this study. Population frequency of variants according to v.3.1.2 (https://gnomad.broadinstitute.org/, accessed on 29 January 2024), for pathogenicity calculation two machine learning models were used, CADD (tool for scoring the deleteriousness of single nucleotide variants and more complex DNA changes; https://cadd.gs.washington.edu/, accessed on 29 January 2024) and funNcion -functional variant prediction in Navs and Cavs ion channels, which also provide predictions for functional changes (https://funnc.shinyapps.io/shinyappweb/, accessed on 29 January 2024). Additional evaluation of variant pathogenicity was performed by analysis of paralogs in *SCNxA* using a tool for missense variant interpretation—PER viewer (https://per.broadinstitute.org/, accessed on 29 January 2024). Functional studies were performed only for a small number of *SCN1A* variants related to identified mutations are mentioned.

| | c.DNA | Protein | Inher. | gnomAD freq. | CADD (1–49) | Paralogous Pathogenic Variant Gene/Variant/Phenotype | Pathogenicity/ Funct.pred. | Functional Studies |
|---|---|---|---|---|---|---|---|---|
| S_1 | 235A>G | Asp79Gly | de novo | 0 | 27.9 | - | Pat/LOF | - |
| S_2 | 241G>A | Asp81Asn | de novo | 0 | 26.3 | *SCN2A*/p.Asp82Gly/ASD | Neut/prob. LOF* | - |
| S_4 | 298T>A | Phe100Ile | pat moz | 0 | 27.1 | - | Neut/prob. LOF* | - |
| S_5 | 302G>A | Arg101Gln | de novo | 0 | 28.5 | - | Pat/LOF | - |
| S_7 | 680T>G | Ile227Ser | de novo | 0 | 30.0 | *SCN8A*/p.Ile231Thr/DEE | Pat/LOF | LOF [13] |
| S_9 | 773T>C | Leu258Pro | nd (ma. -/-) | 0 | 29.0 | - | Pat/GOF | - |
| S_10 | 1025C>T | Ala342Val | de novo | 0 | 32.0 | - | Neut/prob. LOF* | - |
| S_11 | 1247A>G | Asn416Ser | de novo | 0 | 26.1 | - | Pat/GOF | - |
| S_16 | 2585G>A | Arg862Gln | de novo | 0 | 24.1 | *SCN2A*/p.Arg853/DEE | Pat/LOF | - |
| S_17 | 2791C>T | Arg931Cys | de novo | 0 | 32.0 | - | Pat/LOF | - |
| S_18 | 2837G>A | Arg946His | de novo | 0 | 28.1 | *SCN2A*/p.Arg937Cys/His/ASD,Epi *SCN8A*/p.Arg931Gln/NNDwoE/GGE | Pat/LOF | LOF [14] |
| S_22 | 4168G>A | Val1390Met | de novo | 0 | 24.1 | - | Neut/prob. LOF* | - |
| S_24 | 4388T>C | Phe1463Ser | ma.moz | 0 | 29.8 | - | Pat/LOF | - |
| S_26 | 4532T>G | Met1511Arg | nd | 0 | 27.1 | *SCN2A*/p.Met1501Val/DEE *SCN8A*/p.Met1492Thr/UE | Pat/GOF | - |
| S_30 | 4787G>A | Arg1596His | pa. (fam) | $4 \times 10^{-6}$ | 26.1 | - | Neut/prob. LOF* | |
| S_31 | 4786C>T | Arg1596Cys | pa. (fam) | 0 | 29.6 | - | Neut/prob. LOF* | LOF/pLOF [16,17] |
| S_33 | 4964G>T | Gly1655Val | de novo | 0 | 26.1 | - | Pat/LOF | |
| S_34 | 5107G>T | Asp1703Tyr | de novo | 0 | 24.8 | *SCN3A*/p.Asp1688Tyr/UE | Pat/LOF | - |
| S_35 | 5129T>C | Phe1710Ser | de novo | 0 | 29.1 | - | Pat/LOF | - |
| S_37 | 5432T>A | Val1811Asp | nd | 0 | 27.9 | - | Pat/GOF | - |
| S_40 | 5779A>G | Arg1927Gly | de novo | 0 | 27.2 | - | Pat/LOF | LOF [15] |

Pat—pathogenic, Neut.—neutral, LOF—loss of function, prob. LOF—probably loss of function, pLOF—partially LOF, GOF—gain of function, ASD—autism spectrum disorders, DEE—developmental and epileptic encephalopathy, Epi—epilepsy, UE-unclassified epilepsy, GGE—genetic generalized epilepsy, ma., pa.—maternal, paternal inheritance, moz—mosaicism, nd—no data.

## 4. Discussion

From the analysis of the clinical course of the study group, the problem of early diagnosis of DRVT and distinguishing between this and febrile convulsions becomes apparent. In our group, despite the focal nature of the first-ever epileptic seizures, which were present in most of the patients in the study (53%), all were diagnosed with simple febrile seizures, and no extension of neurological diagnosis or continuation of neurological care was suggested. According to the ILAE classification, the correct diagnosis in these patients should be complex febrile seizures (CFS). CFS is defined as having one or more of the following three features: (1) recurrent within 24 h or within the same febrile illness; (2) prolonged duration (>15 min); (3) a focal onset or showing focal features during the seizure. The risk of developing epilepsy if one of the indicators is present ranges from 6% to 8%; with two or three CFS features, the risk is 17% to 22% and 49%, respectively. In the study group, 46% of patients had one risk factor, and 8% had two of them (focal seizures and duration over 15 min). In many authors' opinion, when two or more features of CFS occur, there is an unquestionable need for an *SCN1A* genetic test to confirm the diagnosis of DRVT [18,19]. This indicates the persistent problem of properly classifying seizure types, especially by pediatricians, lifeguards and GPs.

The age of the onset of the first seizure also appears to be important in differentiating DRVT and febrile seizures. In FS and CFS, according to ILAE classification, the typical age is between 6 months and 6 years, but the most frequently observed at 12–18 months [11]. In DRVT, the age of the first symptoms is slightly lower; according to most publications, it is less than 12 months, and the average age varies from 5 to 9 months, depending on the study [20,21]. Our study group had children aged 2–9 months, with a mean age of 5.3. This is below the CFS value, confirming the data above. Also, of significance seems to be the type of first epileptic seizure, especially the occurrence of status epilepticus as the first manifestation of the disorder. According to [10], this is a nonsensitive but highly specific predictive feature of DRVT.

Another valuable parameter for early diagnosis of DRVT is the occurrence of seizures up to 72 h after vaccination. In many studies, the occurrence of these episodes is closely related to the presence of mutations in the *SCN1A* gene [22,23]. In our group, we had to deal with the above phenomenon in 15% of patients.

Intellectual disability was found in 71% of patients. This confirms data from the other publications, where the prevalence of intellectual disability is estimated at around 86–87% [24,25]. It seems that this slightly lower frequency in our study may be due to several factors and limitations. In some of the remaining patients (five patients), the initial level of cognitive functioning was initially expected. However, at longer follow-ups and psychological check-ups, regression of cognitive function and adaptive behavior was evident, with developmental slowing and consequent intellectual impairment, but did not meet the criteria for mild disability. In several of the remaining patients (four patients), the psychological examination was carried out shortly after the diagnosis was made (at the end of the first or beginning of the second year), and they are not currently under the care of our center. We do not observe any significant regression in cognitive development in three patients, who are now 3 years old. These are patients of particular clinical interest because of the rapid diagnosis (in the first year of life) and the therapeutic success since it was possible to reduce the number of epileptic seizures in these patients significantly. All of them were treated with valproic acid and stiripentol, but a remarkable reduction in seizures was obtained after the addition of fenfluramine. Since then, these patients have experienced isolated myoclonic seizures and infrequent tonic–clonic seizures, but only during infections. Perhaps in DRVT, similar to, for example, tuberous sclerosis, rapid inhibition of epileptiform activity can have a significant effect on cognitive regression or autistic symptoms. As confirmed by work on animal models [26], the problem requires further research. Based on the above study and other work, it seems essential to revise the ILAE criteria for DRVT. Patients who meet all the criteria except intellectual disability lose the possibility of effective personalized treatment or qualification for clinical trials. In

the oldest patients, severe intellectual disability is observed more frequently. This may result from a later age of diagnosis and, thus, a delay in the inclusion of effective treatment. However, it may also be due to the very nature of the typical course of DRVT.

In the study group, a diagnosis of autism was clinically confirmed in 18% of patients. Depending on the study, rates of "autistic traits" range from 8.3% to 61.5% [27,28]. It should be noted, however, that many studies consider "autistic traits" without using standardized tools. In our group, as in most studies, impairments in verbal communication were reported to be the greatest, followed by social deficits and less frequent attachment to routines or autostimulation.

The types of epileptic seizures occurring in the study group were typical of DRVT. The most common provocative factor was hyperthermia, which has also been confirmed in many papers [29]. The classical hot water-induced seizures reported in other studies (59% in [30], 22.5% in [31]) were not observed in our study, which is similar to [32]. Perhaps the reason is the detailed psychoeducation of the families of patients with Dravet syndrome, including the need to avoid potential triggers.

In Poland, molecular analysis of the *SCN1A* gene has been available since 2010. This visibly translates into the age of DRVT diagnosis, as shown in Figure 3. The first patients diagnosed were adolescents, had a long history of treatment inadequate for their diagnosis, and frequently received drugs that worsened their clinical condition (sodium channel blockers). Thanks to the increasing awareness of clinicians as well as the greater availability of genetic testing, patients are being diagnosed much earlier, even at one year of age.

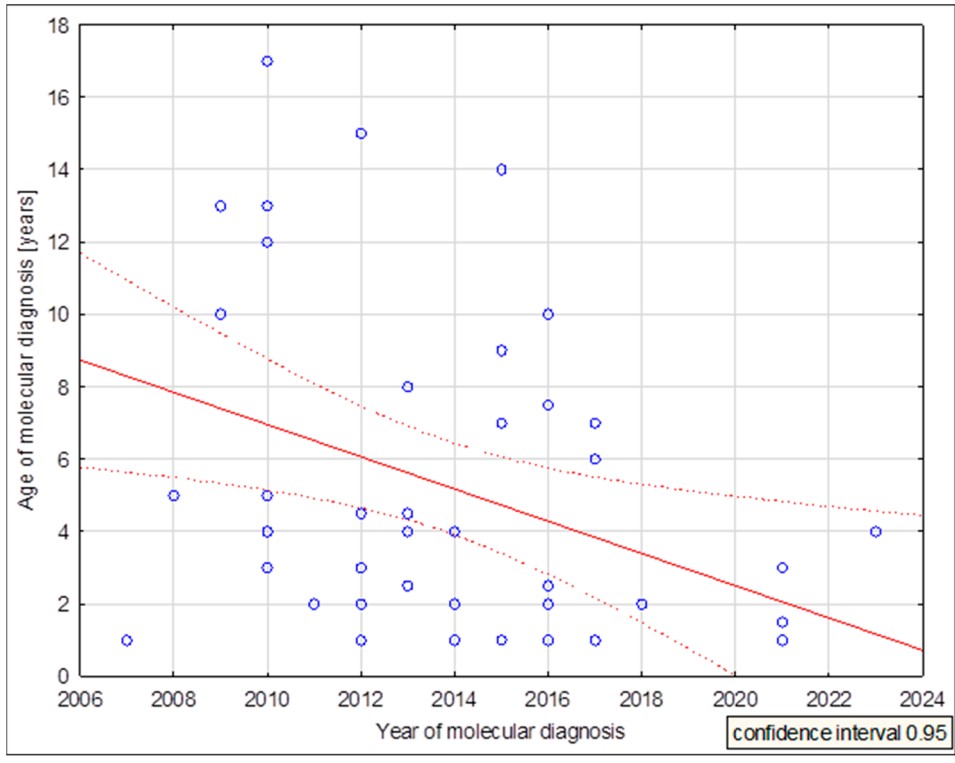

**Figure 3.** DRVT molecular diagnosis age dispersion chart in years vs. patient age. (mean age 5.283, correlation coefficient $-0.375768$, $p = 0.012$).

It is crucial to understand genotype–phenotype associations in *SCN1A*-related epilepsies in order to provide early diagnosis and treatment. In Gallagher's study [10], such correlations have been noted; for example, patients with missense variants in functionally essential regions exhibited earlier seizure onset and were more frequently diagnosed with DRVT. No such correlations were confirmed in our study. This may be mainly due to the small size of our group. The authors [10] emphasize that DRVT does not have strong

genotype–phenotype associations and further studies of cohorts of identical carriers of variants with divergent phenotypes are needed.

## 5. Conclusions

The *SCN1A* gene was first associated with genetic (formerly generalized) epilepsy with febrile seizures plus (GEFSP2/GEFS+) [1]. Subsequently, it was discovered that it is also associated with epileptic encephalopathy—DRVT [33]—and that the vast majority of patients with *SCN1A* variants show this phenotype. Here, we analyzed a cohort of patients diagnosed with DRVT diagnosed and treated within a single medical center. The present study represents the first phenotypic–genotypic analysis of Polish patients with mutations in the *SCN1A* gene. The study focuses on epidemiology, mortality assessment, genetics, seizure characteristics and treatment.

The significant decrease in the age of genetic verification of the diagnosis, as well as the decline in the use of sodium channel inhibitors, underscores the growing attention to the diagnosis of Dravet syndrome among child neurologists in Poland. However, the need for further education of the medical community, particularly pediatricians, on the differentiation of simple and complex febrile seizures and the characteristic symptoms of Dravet syndrome remains an important issue. The distinctive factors of complex febrile convulsions and Dravet syndrome described above may be clinically relevant. This is particularly important in Poland, as pediatricians have the most extensive contact with the patient and decide the need for specialized care.

The first step toward precise medicine is precise diagnosis. Thus, in molecular diagnostics, the correct identification and the characteristics of the pathogenic variant are necessary for the proper management of the patient and the implementation of appropriate treatment. Early and accurate genetic diagnosis is crucial, as it will help the clinician in the appropriate selection of antiepileptic drugs, which can prevent complications and improve the patient's prognosis.

**Author Contributions:** Conceptualization, E.S. and D.H.-Z.; methodology, E.S., D.H.-Z. and P.G.-S.; patients' clinical evaluation and description, E.S., A.Z. and T.M.; sequencing NGS/Sanger, data analysis, and interpretation, P.G.-S., K.K. and R.T.; manuscript preparing, E.S., A.Z. and D.H.-Z.; review and editing, A.Z. All authors have read and agreed to the published version of the manuscript.

**Funding:** This research was funded by the Polish National Science Centre, grant numbers 2015/17/B/NZ4/02669, 800/N-ESF-EuroEPINOMICS/102011/0, NN407054439. The APC was funded by the Director of the Institute of Mother and Child from the funds of the Polish Minister of Education and Science.

**Institutional Review Board Statement:** This study was conducted in accordance with the Declaration of Helsinki and approved by the Institutional Review Board of Institute of Mother and Child at 2009, 2010 and 2016 for studies involving humans (Decision No. 14/2009, 18/2010, 1/2016).

**Informed Consent Statement:** Informed consent was obtained from all subjects involved in the study.

**Data Availability Statement:** The data presented in this study are available upon request from the corresponding author.

**Conflicts of Interest:** The authors declare no conflicts of interest.

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
