# Peer review of "SCN1A—Characterization of the Gene’s Variants in the Polish Cohort of Patients with Dravet Syndrome: One Center Experience"

_cimb, doi:10.3390/cimb46050269_

Round 1

Reviewer 1 Report

Comments and Suggestions for Authors

To the AA

Aim of the present study was the characterization of genotype and phenotype heterogeneity in pts with SCN1A mutations and Dravet Syndrome (DRVT) in a retrospective, single centre-study (n=50 pts). No genotype-phenotype relationship was evidenced. Co-occurrence of intellectual disability ( 71%) and ASD  (18%) was reported, and the difficulty in discriminating between febrile convulsions (FC) and DRVT was observed. The AA conclude from their analysis that age at the first seizure (< 1 year of age), onset of a seizure up to 72 hours after vaccination and the presence of more than two features of complex febrile seizures would indicate DRVT, thus leading to a correct diagnosis and a personalized treatment.

The Ms is well organized and well documented. The molecular features of the DRVT pts are well described. Main limitations are single-centre study and retrospective design.

Minor Points

1. Treatment is not sufficiently reported. Since personalized treatment of DRVT is of paramount relevance in the clinical management, I would encourage the AA to include these data.

2. Although the Ms. is generally well-written, a moderate language revision is needed. 

Comments on the Quality of English Language

Although the Ms. is generally well-written, a moderate language revision is needed.

Author Response

We do not have accurate data on the course of treatment in patients. 
Patients after receiving a diagnosis of Dravet syndrome mostly had a treatment regimen dedicated to Dravet syndrome, i.e. valproic acid, clobazam and stiripentol. Prior to the diagnosis, treatment with sodium channel blockers was in several patients with significant incrising of seizures. 

Reviewer 2 Report

Comments and Suggestions for Authors

The manuscript presents a study of the first phenotypic-genotypic analysis of Polish patients with mutations in the SCN1A gene.

The work contains data on genetic and pathogenetic characteristics of the variants identified in 50 DRVT patients; characteristic of SCN1A gene missense variants identified among patients.

In the discussion section, the authors discussed the results obtained in detail and compared them with already known and published data.

The practical recommendation of this work is to carry out early diagnosis to prescribe optimal treatment.

There are several comments about the work:

1. The abstract must be corrected in accordance with the requirements of the journal.

2. The introduction section contains only 5 references to literary sources. Typically, the introduction section is an overview of the current state of the scientific problem; it should contain a number of links to articles by other authors or to your own articles in this area of research. The review should end with a statement of the problem. In the review, each idea must be supported by a reference to a literary source.

3. In the materials and methods section there is no information about what equipment the sequencing was used on.

4. In the conclusion section, the last paragraph is obvious. It contains truths that are absolutely understandable to everyone. It needs to be redone or removed.

Author Response

1. We have improved. 
2  We have added more sources.
3 We have added this information.
4) We have removed part of the paragraph.  
Thank you for your review and guidance

Round 2

Reviewer 2 Report

Comments and Suggestions for Authors In this version of the manuscript, the authors have made the necessary corrections. I believe that in this form the article can be accepted for publication.

Regards,